# *DNMT3B* rs2424913 as a Risk Factor for Congenital Heart Defects in Down Syndrome

**DOI:** 10.3390/genes14030576

**Published:** 2023-02-24

**Authors:** Dijana Majstorović, Anita Barišić, Ivana Babić Božović, Iva Bilić Čače, Neven Čače, Mauro Štifanić, Jadranka Vraneković

**Affiliations:** 1Faculty of Medicine, Juraj Dobrila University of Pula, 52100 Pula, Croatia; 2Faculty of medicine, Department of Medical Biology and Genetics, University of Rijeka, 51000 Rijeka, Croatia; 3Clinical Institute of Genomic Medicine, University Medical Centre Ljubljana, 1000 Ljubljana, Slovenia; 4Clinical Hospital Centre Rijeka, Department of Pediatrics, 51000 Rijeka, Croatia; 5Faculty of Natural Sciences, Juraj Dobrila University of Pula, 52100 Pula, Croatia

**Keywords:** congenital heart defect, DNA methyltransferase, Down syndrome, *MTHFR*, *MTRR*, single-nucleotide polymorphism

## Abstract

Impairments of the genes that encode enzymes that are involved in one-carbon metabolism because of the presence of gene polymorphisms can affect the methylation pattern. The altered methylation profiles of the genes involved in cardiogenesis may result in congenital heart defects (CHDs). The aim of this study was to investigate the association between the *MTHFR* rs1801133, *MTHFR* rs1801131, *MTRR* rs1801394, *DNMT1* rs2228611, *DNMT3A* rs1550117, *DNMT3B* rs1569686, and *DNMT3B* rs2424913 gene polymorphisms and congenital heart defects in Down syndrome (DS) individuals. The study was conducted on 350 participants, including 134 DS individuals with CHDs (DSCHD+), 124 DS individuals without CHDs (DSCHD−), and 92 individuals with non-syndromic CHD. The genotyping was performed using the PCR–RFLP method. A statistically significant higher frequency of the *DNMT3B* rs2424913 TT in the DSCHD+ individuals was observed. The *DNMT3B* rs2424913 TT genotype, as well as the T allele, had significantly higher frequencies in the individuals with DS and atrial septal defects (ASDs) in comparison with the individuals with DS and other CHDs. Furthermore, our results indicate a statistically significant effect of the *DNMT3B* rs1569686 TT genotype in individuals with non-syndromic CHDs. The results of the study suggest that the *DNMT3B* rs2424913 TT genotypes may be a possible predisposing factor for CHDs in DS individuals, and especially those with ASDs.

## 1. Introduction

Congenital heart defects (CHDs) are the most common type of congenital malformations, with an incidence of 17.9/1000 [1], and they cause approximately 300,000 infant deaths per year [2]. CHDs are defined as a developmentally and clinically heterogeneous group of structural and functional congenital malformations of the heart and/or large blood vessels that result from the incomplete development of the heart during cardiac embryogenesis [3]. Chromosomal aberrations contribute to 10–12% of all CHD cases in live births [4]. In Down syndrome (DS), which is the most common chromosomal abnormality worldwide and is caused by a trisomy of human chromosome 21, CHDs are present in 40–50% of all cases [5]. The most common CHDs in individuals with DS are atrioventricular septal defects (AVSDs) (51%), ventricular septal defects (VSDs) (25%), atrial septal defects (ASDs) (9%), and tetralogy of Fallot (TF) (7%) [6]. The exact etiology that underlies CHDs is still unknown. A structurally normal heart is present in more than half of DS individuals; therefore, CHDs are not exclusively a consequence of trisomy 21 [7,8,9]. Two hypotheses have been proposed to explain DS-associated CHDs. According to the gene dosage hypothesis, CHDs could be the result of the increased expression of certain genes on chromosome 21 [10,11]. The gene mutation hypothesis states that various locus-specific mutations may be the underlying mechanism in the development of CHDs [7,11]. Today, we know that the development of CHDs requires the interaction of various genetic and epigenetic risk factors. Therefore, several risk factors have been proposed, such as locus-specific mutations, numerous single-nucleotide polymorphisms (SNPs), copy number variations (CNVs), microRNAs (miRNA), and various signaling and metabolic pathways [12,13]. One of the proposed risk factors is the carbon metabolic pathway, which includes the folate acid and homocysteine (Hcy) pathways. These metabolic pathways provide methyl groups for the cellular processes of nucleotide synthesis and methylation [14]. Methylenetetrahydrofolate reductase (MTHFR) is one of the key regulatory enzymes in the folate pathway. This enzyme catalyzes the conversion of methylenetetrahydrofolate to the main form of circulating folate, 5-methyltetrahydrofolate (5-metylTHF), which is required for the remethylation of Hcy to methionine. Methionine synthase reductase (MTRR) is another important enzyme for the conversion of the inactive form of methionine synthase to the active form, and it contributes to the transfer of the methyl group [15]. The methyl group is essential for DNA methylation and is the epigenetic mechanism involved in the regulation of many critical cellular processes during human development and throughout life, such as transcriptional regulation, genomic imprinting, chromosome maintenance, and genomic stability [16]. A number of studies suggest that aberrant DNA methylation may play an important role in many diseases [17]. The DNA methylation pattern is established and maintained by three DNA methyltransferases (DNMTs), with the addition of a methyl group to the 5′ position of the cytosine at a CpG site [18]. DNA methyltransferase 1 (DNMT1) authentically copies methylation information from parent strands to daughter strands during DNA replication. Because of this role, DNMT1 is referred to as a “maintenance” methyltransferase. Two other DNA methyltransferases, DNMT3A and DNMT3B, have de novo DNMT catalytic activities to establish methylation patterns during early embryogenesis [19]. An overview of the folate metabolic pathway is shown in the schematic in Figure 1 [20].

The development of the heart is a complex process. DNA methylation plays a crucial role in the expression of cardiac genes during development [21]. Cardiac DNA methylation varies during life, and it is different in the embryonic stages, between neonates and adults, and between healthy individuals and patients [22]. Genetic variation caused by SNPs is the most common form of change in the gene structure, and it may influence gene expression. Literature evidence suggests that polymorphisms of the *MTHFR*, *MTRR,* and *DNMTs* genes could be a risk factor for CHDs; however, the results are inconsistent [23,24,25]. Specifically, the *MTHFR* 677C>T (rs1801133) polymorphism is a common genetic variation that affects the function of the MTHFR enzyme. The *MTHFR* 677TT genotype is associated with a decrease in MTHFR enzyme activity of about 70% compared with the wild-type CC genotype. The CT genotype is associated with a reduction in enzyme activity of about 40%. As a result, individuals with the TT or CT genotype may have lower levels of 5-methyl-THF and higher levels of homocysteine in their blood. Likewise, the *MTHFR* 1298A>C (rs1801131) polymorphism is another common variation in the *MTHFR* gene that affects enzyme activity. The *MTHFR* 1298CC genotype is associated with a reduction in MTHFR enzyme activity of about 30% compared with the wild-type AA genotype. The AC genotype is associated with a smaller reduction in enzyme activity of about 15% [26]. The most common polymorphism of the *MTRR* gene is *MTRR* 66A>G (rs1801394). Studies have shown that the *MTRR* 66A>G polymorphism leads to decreased MTRR enzyme activity and reduced efficiency in the remethylation of homocysteine to methionine [27]. The most studied *DNMT1* SNPs are 97T>C (rs16999593), +32204A>G (rs2228611), and 327A>G (rs2228612), which exist in the coding regions and, consequently, may influence the *DNMT1* expression [23,28,29]. Two common polymorphisms have been identified in the promoter region of the *DNMT3B* gene: −149C>T (rs2424913) and −579G>T (rs1569686). The *DNMT3B* −149C>T polymorphism is known to have an impact on *DNMT3B* gene expression [30]. The *DNMT3A* −448A>G (rs1550117) polymorphism is also present in the promoter region. The altered expressions of *MTHFR*, *MTRR,* and *DNMTs* can result in changes in DNA methylation patterns, which can have downstream effects on gene expressions relevant to heart development and function.

Because the *MTHFR, MTRR*, and *DNMT* genes are critical for maintaining DNA methylation patterns, we hypothesized that the polymorphisms in these genes might be the risk factors for CHDs in individuals with trisomy 21. Therefore, we selected the SNPs considering their influence on methylation during development and later in life, and we conducted a study to investigate the association between the *MTHFR* rs1801133, *MTHFR* rs1801131, *MTRR* rs1801394, *DNMT1* rs2228611, *DNMT3A* rs1550117, *DNMT3B* rs1569686, and *DNMT3B* rs2424913 gene polymorphisms and CHDs in DS individuals and controls.

## 2. Materials and Methods

### 2.1. Patients

This study was conducted on 350 participants, including 134 DS individuals with CHDs (DSCHD+), 124 DS individuals without CHDs (DSCHD−), and 92 individuals with non-syndromic CHDs (Table 1). The study population was of the same race (Caucasian). Participants with DSCHD+ and DSCHD− were selected from the Clinical Hospital Center Rijeka in collaboration with DS associations in Croatia. Participants with non-syndromic CHDs were selected from the Clinical Hospital Center Rijeka, the Pula General Hospital, and Istrian health centers. Trisomy 21 was confirmed by karyotyping, and CHDs were confirmed by ultrasound. Written informed consent was obtained from all participants (for minors, the consent was obtained from their legal guardians). The study protocol was approved by the Institutional Review Board of the Ethics Committee for Biomedical Research of the Faculty of Medicine (ethical approval number: 003-08/19-01/114; 2170-24-09-8-19-2), the University of Rijeka, the Clinical Hospital Center Rijeka, the Pula General Hospital, and Istrian health centers. The study was conducted in accordance with the Declaration of Helsinki.

### 2.2. DNA Isolation and Genotyping

Genomic DNA was isolated from peripheral blood leukocytes and buccal epithelial cells according to a standard procedure and using a commercially available kit (Qiagen FlexiGene^®^ DNA Kit (250), Qiagen GmbH, Hilden, Germany and QIAamp^®^ DNA Mini Kit (50), Qiagen GmbH, Hilden, Germany), and it was stored at −20 °C. Buccal epithelial cells were collected from participants who did not consent to blood sampling. The concentration of DNA was above 50 ng/μL. The A260/A280 ranged from 1.8 to 2.0. *MTHFR* 677C>T (rs1801133) and *MTHFR* 1298A>C (rs1801131), *MTRR* 66A>G (rs1801394), *DNMT1* +32204A/G (rs2228611), *DNMT3A* −448A/G (rs1550117), *DNMT3B* −579G/T (rs1569686), and *DNMT3B* −149C/T (rs2424913) were analyzed using polymerase chain reaction–restriction fragment length polymorphism (PCR–RFLP) (Appendix A). To assess the reproducibility of the results, the samples were run in duplicate, and at the end, 10% of all samples were randomly repeated. DNA samples of the wild-type and mutant homozygous genotypes were used as quality control samples in each enzyme digestion reaction.

### 2.3. Statistical Analysis

Statistical analysis was performed with Statistica for Windows, version 13.3 (StatSoft, Inc., Tulsa, OK, USA), and with MedCalc for Windows, version 14.12.0. The normality of the distribution was tested with the Kolmogorov–Smirnov test. A variable that was not normally distributed was expressed by the median and range. Nominal indicators were represented by frequency distributions by groups and proportion. A Pearson chi-square test and Fischer exact test were used to determine the differences in genotype and allele frequencies and the deviation from the Hardy–Weinberg equilibrium between groups. Odds ratios (ORs) with 95% confidence intervals (95% CIs) were estimated according to dominant, recessive, and codominant models for the analysis of the genetic association of genotypes and alleles. Statistical significance was considered at *p* ≤ 0.05.

## 3. Results

Table 1 shows the demographic characteristics of the study populations. All the genotype frequencies were consistent with Hardy–Weinberg expectations. Statistically significant higher frequencies of *DNMT3B* rs2424913 CT (χ^2^ = 5.37; *p* = 0.021) in the DSCHD− and TT (χ^2^ = 5.63; *p* = 0.019) in the DSCHD+ were observed (Table 2). No statistically significant associations were found for the other analyzed gene polymorphisms (Table 2). A significant risk for CHDs according to the dominant (CC + CT vs. TT) and codominant genetic (TT vs. CT) models was observed for *DNMT3B* rs2424913 in the DSCHD+ and DSCHD− groups (OR = 2.24; 95% CI = 1.13–4.42; *p* = 0.019; OR = 0.38; 95% CI = 0.18–0.77; *p* = 0.008, respectively) (Table 3). Statistically significant effects of the *DNMT3B* rs1569686 TT genotype (χ^2^ = 4.37; *p* = 0.039) (Table 2), as well as of the dominant (GG + TG vs. TT) (OR = 2.08; 95% CI = 1.03–4.20; *p* = 0.039) and codominant (TT vs. TG) genetic models (OR = 0.46; 95% CI = 0.22–0.98; *p* = 0.046) in the individuals with non-syndromic CHDs was observed (Table 3). The *DNMT3B* rs2424913 TT genotype and T allele had significantly higher frequencies in the DS individuals with ASDs compared with the DS individuals with other CHDs (χ^2^ = 4.97; *p* = 0.028; χ^2^ = 5.69; *p* = 0.018, respectively) (Table 4). The most frequent congenital heart defects in the DS and non-syndromic individuals were ASDs, followed by VSDs and AVSDs, with no difference in the sex distribution.

## 4. Discussion

To the best of our knowledge, this is the first study to assess the association between *MTHFR*, *MTRR*, and *DNMT* gene polymorphisms and CHDs in individuals with DS and individuals with non-syndromic CHDs. The results of our study indicate that the *DNMT3B* rs2424913 TT genotype and dominant and codominant genetic models of the same polymorphism may be potential predisposing factors for CHDs in individuals with DS, and particularly in individuals with ASDs. Furthermore, our results suggest that the *DNMT3B* rs1569686 TT genotype increased the CHD risk in the individuals with non-syndromic CHDs compared with the DS group with CHDs, which was also found under the dominant and codominant genetic models of the same polymorphism.

In the presence of the *DNMT3B* rs2424913 polymorphism, the expression of the *DNMT3B* gene is changed. The minor T allele increases the promoter activity by 30% [31,32] and affects the miRNA binding site [33]. Considering this, we propose that the increased *DNMT* gene expression due to the presence of the T allele of the rs2424913 SNP alters the DNA methylation of the relevant genes in cardiogenesis and potentially leads to CHDs. Moreover, this gene may be an additional risk factor for the altered methylation profiles of the genes on chromosome 21 (*DSCAM, COL6A1, COL6A2, KCNJ6*, *RCAN1*) related to the endocardial cushion type in DSCHD+, as suggested in a recent review [12,13]. In our study, we also found a significant risk of the *DNMT3B* rs1569686 TT genotype and dominant and codominant genetic models of the same polymorphism for CHDs in individuals with non-syndromic CHDs. The functional role of this polymorphism has not yet been clearly defined. Altered gene splicing, linkage disequilibrium with other polymorphisms, and the deactivation of illegitimate transcription factor binding are possible mechanisms by which rs1569686 polymorphisms affect gene expression [31,33,34].

Moreover, animal experiments have shown that one of the most active enzymes is *Dnmt3b*, which is strongly expressed in the early embryonic stages in DNA methylation reprogramming during cardiomyocyte differentiation [35,36,37]. The knockdown of Dnmt3b in mice leads to death due to ventricular defects [37]. Moreover, Chamberlain et al. found that Dnmt3b regulates the DNA methylation of the cardiac essential genes and, in combination with Dnmt3a, causes the repression of fetal cardiac genes [38]. Furthermore, a study by Zhang et al. has shown that approximately 60% of the RE1 silencing transcription factor target genes (REST target genes) in cardiac embryogenesis are at least partially controlled by Dnmt3b [39]. Interestingly, according to the methylation quantitative trait loci (mQTL) database, *DNMT3B* rs2424913 and rs1569686 are also the mQTLs that are active throughout the lifespan, and they are located in close proximity to multiple DNA methylation sites [40]. Although we did not find a significant association between *DNMT3A* rs1550117 and CHDs in syndromic or non-syndromic individuals, further studies are needed to clarify its role. Indeed, other studies have demonstrated correlations between *DNMT3A* and the tetralogy of Fallot [30] and cardiac fibrosis [41]. Moreover, previous *Dnmt3a* knockdown studies in animals revealed that Dnmt3a plays an important role in heart development by regulating embryonic cardiomyocyte gene expression, morphology, and function [36]. In contrast to the de novo catalytic activity of DNMT3A and DNMT3B, DNMT1 is responsible for maintaining methylation patterns [19]. Several previous studies have shown that *DNMT1* polymorphisms (rs16999593, rs2228612) are associated with the risk of developing CHDs [23], and that the decreased expression of DNMT1 may play an important role in the pathogenesis of tetralogy of Fallot [30]. Researchers have also suggested that *DNMT1* rs16999593 reduces the risk of the transposition of the great arteries [29]. Studies in animal models indicate that *Dnmt1* is involved in embryonic cardiomyocytes by regulating DNA methylation, gene expression, gene splicing, and the cell’s function [42].

Although relationships between *DNMT1* rs2228611 and CHDs were not found in our study, DNMT1 is still a good candidate for future investigation, considering that it plays an important role during cardiomyocyte development.

For the *MTHFR* rs1801131 and rs1801133 and *MTRR* rs1801394 polymorphisms, we did not find an association with the CHD risk in the case and control groups. Conflicting results have been shown in previous studies in which the associations of these polymorphisms and CHDs in the general population were investigated, as well as in the population with DS [24,25,27,43,44,45,46,47,48,49]. However, given the known associations of these polymorphisms with decreased enzyme activity, which may consequently lead to an increase in homocysteine levels and thus affect the methylation pattern [22], they still represent good candidates for the development of CHDs and should not be omitted in future studies. The potential limitations of our study include the small sample size and, especially, the subgroup analysis of specific defects. Furthermore, additional research should be focused on the possible influence of gene–gene and gene–environment interactions, considering the multiple etiologies of CHDs.

## 5. Conclusions

We examined the association between the *MTHFR*, *MTRR*, and *DNMTs* gene polymorphisms and CHDs in syndromic and non-syndromic individuals. We found that the *DNMT3B* rs2424913 TT genotypes and the genetic model CC + CT vs. TT could be predisposing factors for CHDs in individuals with Down syndrome and particularly those with ASDs. In addition, our results indicate that the *DNMT3B* rs1569686 TT genotypes increased the risk of CHDs in individuals with non-syndromic CHDs. Accordingly, these findings are noteworthy because they reveal new aspects of CHD etiology. Due to the small sample size of the research, further studies with larger cohorts focused on specific types of CHDs are needed to define the functional roles of the *MTHFR*, *MTRR*, and *DNMTs* gene polymorphisms in CHD etiology.

## Figures and Tables

**Figure 1 genes-14-00576-f001:**
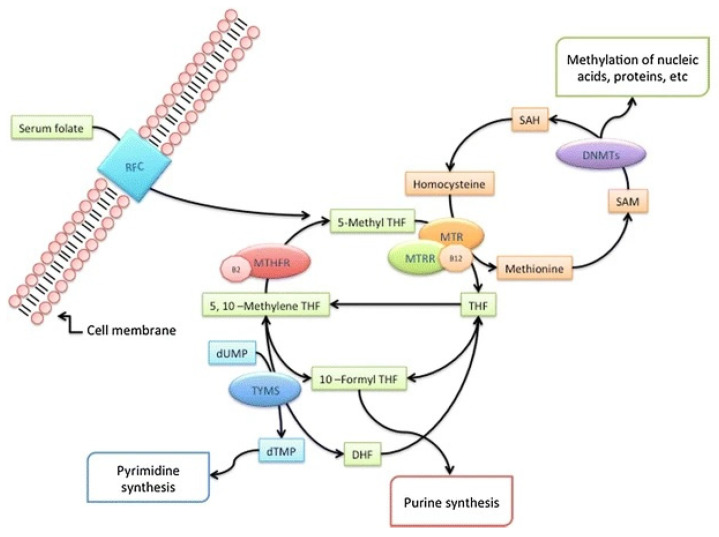
Overview of the folate metabolic pathway (adapted from Coppedè 2015 [20] RightsLink No. 5490861127574). Dietary folates provide one-carbon units for DNA synthesis and methylation reactions. Enzymes: RFC1: reduced folate carrier; DNMT: DNA methyltransferases; MTR: methionine synthase; MTRR: methionine synthase reductase; MTHFR: methylenetetrahydrofolate reductase; TYMS: thymidylate synthase. Cofactors: B2: vitamin B2; B12: vitamin B12. Metabolites: SAH: S-adenosylhomocysteine; SAM: S-adenosylmethionine; 5,10-MTHF: 5,10 methylenetetrahydrofolate; THF: tetrahydrofolate; DHF: dihydrofolate; 10-formyl-THF: 10-formyl-tetrahydrofolate; 5-MTHF: 5-methyltetrahydrofolate; dUMP: deoxyuridine monophosphate; dTMP: deoxythymidine monophosphate.

**Table 1 genes-14-00576-t001:** Characteristics of the study populations.

	DSCHD+	DSCHD−	CHD
No. of cases (%)	134	124	92
Gender			
Male	69 () (51.5)	71 () (57.3)	55 (59.8)
Female	65 () (48.5)	53 () (42.7)	37 (40.2)
Age	Median [range]		
	2 [0–27]	2.5 [0–55]	7.5 [0–32]

DSCHD+: Down syndrome individuals with congenital heart defects; DSCHD−: Down syndrome individuals without congenital heart defects; CHD: individuals with non-syndromic congenital heart defects.

**Table 2 genes-14-00576-t002:** Genotype and allele frequencies of DNA methyltransferases (*DNMTs*), methylenetetrahydrofolate reductase (*MTHFR*), and 5-methyltetrahydrofolate-homocysteine methyltransferase reductase (*MTRR*) gene polymorphisms in study groups.

				DSCHD+ N (%)	DSCHD− N (%)	CHD N (%)	χ^2^ DSCHD+ DSCHD−	*p* DSCHD+ DSCHD−	χ^2^ DSCHD+ CHD	*p* DSCHD+ CHD
*DNMT1*	rs2228611	Genotype	AA AG GG	31 (24.0) 80 (62.0) 18 (14.0)	38 (31.7) 66 (55.0) 16 (13.3)	27 (29.3) 46 (50.0) 19 (20.7)	1.81 1.26 0.02	0.180 0.262 0.887	0.78 3.16 1.73	0.377 0.076 0.191
		Allele	A G	142 (55.0) 116 (45.0)	142 (59.2) 98 (40.8)	100 (54.3) 84 (45.7)	0.87	0.353	0.02	0.886
*DNMT3A*	rs1550117	Genotype	AA AG GG	1 (0.8) 21 (16.7) 104 (82.5)	1 (0.8) 17 (14.3) 101 (84.9)	1 (1.1) 15 (16.5) 75 (82.4)	0.00 0.27 0.24	0.737 0.607 0.622	0.05 0.00 0.00	0.664 0.972 0.981
		Allele	A G	23 (10.0) 229 (90.0)	19 (8.0) 219 (92.0)	17 (9.3) 165 (90.7)	0.20	0.652	0.04	0.485
*DNMT3B*	rs1569686	Genotype	GG TG TT	50 (38.5) 63 (48.5) 17 (13.0)	45 (36.3) 65 (52.4) 14 (11.3)	32 (34.8) 38 (41.3) 22 (23.9)	0.13 0.40 0.19	0.721 0.528 0.664	0.31 1.11 4.37	0.576 0.292 0.039
		Allele	G T	163 (63) 97 (37)	155 (63) 93 (38)	102 (55) 82 (45)	0.00	0.964	2.36	0.125
	rs2424913	Genotype	CC CT TT	41 (33.1) 52 (41.9) 31 (25.0)	35 (30.2) 66 (56.9) 15 (12.9)	29 (32.2) 37 (41.1) 24 (26.7)	0.23 5.37 5.63	0.630 0.021 0.019	0.02 0.01 0.08	0.897 0.904 0.783
		Allele	C T	134 (54) 114 (46)	136 (59) 96 (41)	95 (53) 85 (47)	1.03	0.311	0.07	0.246
*MTHFR*	rs1801133	Genotype	CC CT TT	51 (38.1) 68 (50.7) 15 (11.2)	58 (46.7) 55 (44.4) 11 (8.9)	45 (48.5) 37 (40.2) 10 (10.9)	2.01 1.06 0.38	0.157 0.305 0.537	2.64 2.43 0.01	0.106 0.120 0.939
		Allele	C T	170 (63.4) 98 (36.6)	171 (68.9) 77 (31.1)	127 (69.0) 57 (31.0)	1.75	0.186	1.51	0.219
	rs1801131	Genotype	AA AC CC	76 (56.7) 55 (41.1) 3 (2.2)	55 (44.4) 65 (52.4) 4 (3.2)	41 (44.6) 46 (50.0) 5 (5.4)	3.94 3.35 0.24	0.048 0.068 0.457	3.23 1.77 1.63	0.073 0.184 0.181
		Allele	A C	207 (77.2) 61 (22.8)	175 (70.6) 73 (29.4)	128 (69.6) 56 (30.4)	2.99	0.085	3.35	0.068
*MTRR*	rs1801394	Genotype	AA AG GG	32 (24.1) 70 (52.6) 31 (23.3)	23 (18.5) 72 (58.1) 29 (23.4)	16 (17.4) 59 (64.1) 17 (18.5)	1.09 0.77 0.00	0.297 0.382 0.988	1.44 2.94 0.76	0.232 0.087 0.386
		Allele	A G	134 (50.4) 132 (49.6)	118 (47.6) 130 (52.4)	91 (49.5) 93 (50.5)	0.40	0.527	0.04	0.848

DSCHD+: Down syndrome individuals with congenital heart defects; DSCHD−: Down syndrome individuals without congenital heart defects; CHD: individuals with non-syndromic congenital heart defects; χ^2^: chi-square test; *p*: *p*-value.

**Table 3 genes-14-00576-t003:** Risk for congenital heart defects according to different genetic models in study groups.

Genetic Model		DSCHD+ vs. DSCHD−		DSCHD+ vs. CHD	
		OR (95% CI)	*p*	OR (95% CI)	*p*
*DNMT1* rs2228611					
Dominant	AA + AG vs. GG	1.05 (0.51–2.17)	0.886	1.60 (0.78–3.26)	0.191
Recessive	AA vs. AG + GG	1.46 (0.83–2.55)	0.179	0.76 (0.41–1.39)	0.376
Codominant	AA vs. GG	1.37 (0.60–3.14)	0.444	1.21 (0.53–2.76)	0.648
Codominant	AA vs. AG	1.48 (0.83–2.64)	0.177	0.66 (0.35–1.24)	0.196
Codominant	GG vs. AG	1.07 (0.50–2.27)	0.845	0.54 (0.26–1.14)	0.107
*DNMT3A* rs1550117					
Dominant	AA + AG vs. GG	0.84 (0.42–1.66)	0.621	0.99 (0.48–2.01)	0.981
Recessive	AA vs. AG + GG	1.05 (0.06–17.13)	0.736	0.72 (0.04–11.66)	0.664
Codominant	AA vs. GG	1.02 (0.06–16.68)	0.743	0.72 (0.04–11.71)	0.664
Codominant	AA vs. AG	1.23 (0.07–21.24)	0.703	0.71 (0.04–12.34)	0.671
Codominant	GG vs. AG	1.19 (0.59–2.40)	0.608	0.99 (0.47–2.04)	0.979
*DNMT3B* rs1569686					
Dominant	GG + TG vs. TT	1.18 (0.55–2.51)	0.664	2.08 (1.03–4.20)	0.039
Recessive	GG vs. TG + TT	0.91 (0.54–1.51)	0.720	1.17 (0.67–2.04)	0.576
Codominant	GG vs. TT	1.09 (0.48–2.46)	0.830	2.02 (0.93–4.38)	0.074
Codominant	GG vs. TG	0.87 (0.51–1.48)	0.614	0.94 (0.51–1.71)	0.846
Codominant	TT vs. TG	0.79 (0.36–1.75)	0.574	0.46 (0.22–0.98)	0.046
*DNMT3B* rs2424913					
Dominant	CC + CT vs. TT	2.24 (1.13–4.42)	0.019	1.09 (0.58–2.02)	0.783
Recessive	CC vs. CT + TT	0.87 (0.50–1.50)	0.630	1.20 (0.68–2.14)	0.514
Codominant	CC vs. TT	1.76 (0.82–3.78)	0.145	1.09 (0.53–2.23)	0.804
Codominant	CC vs. CT	0.67 (0.37–1.20)	0.179	1.00 (0.53–1.89)	0.985
Codominant	TT vs. CT	0.38 (0.18–0.77)	0.008	0.91 (0.46–1.81)	0.807
*MTHFR* rs1801133					
Dominant	CC + CT vs. TT	1.29 (0.57–2.93)	0.536	0.96 (0.41–2.25)	0.939
Recessive	CC vs. CT + TT	1.43 (0.87–2.34)	0.157	0.64 (0.37–1.09)	0.105
Codominant	CC vs. TT	1.55 (0.65–3.68)	0.319	0.75 (0.30–1.84)	0.539
Codominant	CC vs. CT	1.40 (0.83–2.35)	0.196	0.61 (0.34–1.08)	0.094
Codominant	TT vs. CT	0.90 (0.38–2.13)	0.822	0.81 (0.33–1.99)	0.656
*MTHFR* rs1801131					
Dominant	AA + AC vs. CC	0.68 (0.15–3.13)	0.457	2.50 (0.58–10.77)	0.180
Recessive	AA vs. AC + CC	0.60 (0.37–0.99)	0.047	1.62 (0.95–2.78)	0.073
Codominant	AA vs. CC	0.54 (0.11–2.52)	0.341	3.08 (0.70–13.58)	0.120
Codominant	AA vs. AC	0.61 (0.37–1.00)	0.054	1.55 (0.89–2.67)	0.115
Codominant	CC vs. AC	1.12 (0.24–5.25)	0.596	0.50 (0.11–2.21)	0.288
*MTRR* rs1801394					
Dominant	AA + AG vs. GG	0.99 (0.55–1.77)	0.988	0.74 (0.38–1.44)	0.385
Recessive	AA vs. AG + GG	0.71 (0.39–1.31)	0.282	1.50 (0.77–2.94)	0.231
Codominant	AA vs.GG	0.76 (0.36–1.60)	0.483	1.09 (0.47–2.54)	0.829
Codominant	AA vs. AG	0.69 (0.37–1.31)	0.263	1.68 (0.84–3.37)	0.139
Codominant	GG vs. AG	0.90 (0.49–1.66)	0.758	1.53 (0.77–3.05)	0.219

DSCHD+: Down syndrome individuals with congenital heart defects; DSCHD−: Down syndrome individuals without congenital heart defects; CHD: individuals with non-syndromic congenital heart defects; OR: odds ratio; CL: 95% confidence interval; *p*: *p*-value.

**Table 4 genes-14-00576-t004:** Genotype and allele frequencies of *DNMT3B* rs2424913 gene polymorphism in Down syndrome individuals with atrial septal defects (ASDs) and other congenital heart defects (CHDs).

	ASD N (%)	CHD N (%)	χ^2^	OR (95% Cl)	*p*-Value
CC	11 (24.4)	30 (38.0)	2.37	0.53 (0.23–1.20)	0.126
CT	18 (40.0)	35 (44.3)	0.22	0.84 (0.40–1.76)	0.642
TT	16 (35.6)	14 (17.7)	4.97	2.56 (1.11–5.93)	0.028
CC vs. CT + TT	Recessive		2.37	0.53 (0.23–1.20)	0.126
CC + CT vs. TT	Dominant		4.97	0.39 (0.17–0.90)	0.028
Allele			5.69	0.53 (0.31–0.90)	0.018
C	40 (44.4)	95 (60.1)			
T	50 (55.6)	63 (39.9)			

χ^2^: chi-square test; OR: odds ratio; Cl: confidence interval.

## Data Availability

All data from this research are available from the corresponding author upon reasonable request.

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
