# Peer review of "DNMT3B rs2424913 as a Risk Factor for Congenital Heart Defects in Down Syndrome"

_genes, 2023, doi:10.3390/genes14030576_

Round 1

Reviewer 1 Report

Reviewer comment: the current manuscript aim to investigate the association between MTHFR, MTRR and DNMT gene polymorphisms and congenital heart defects (CHDs) in Down syndrome (DS) individuals. The results suggest that only DNMT3B rs2424913 TT genotypes may be a possible predisposing factor for CHDs in DS patients, especially those with atrial septal defect, revealing new aspects of CHD etiology.

The authors took up an interesting topic of research and  the results are clearly reported. The English form is linear and flowing.

There are only few minor comments:

1) I suggest the authors to add a schematic figure of folate/methionine/homocysteine ​​metabolism with related enzymes to better understand the metabolic pathway which is well explained in the introductory section.

2) Do you check any difference among groups in gender and age? CHD group in Table 1 seems to have higher median age compared to DS groups (both CHD+ and CHD-).

3) Reference 3. The first part of the title is missing.

4) The table numbers in the text are wrong: line 166, you have to correct "Table 1" with "Table 3" and line 171 "Table 2" with "Table 4".

5) Line 144-145. “Analyzed” is reported twice.

Author Response

Response to Reviewer 1 Comments

We would like to thank the reviewer for valuable comments and observations, which improved our paper. We appreciate the time and effort spent in analyzing our paper. Please find detailed response below. All changes in the manuscript are marked up using the “Track Changes” function.

Point 1: I suggest the authors to add a schematic figure of folate/methionine/homocysteine ​​metabolism with related enzymes to better understand the metabolic pathway which is well explained in the introductory section.

Response 1: we add a schematic figure of folate/methionine/homocysteine metabolism with related enzymes, as recommended.

Point 2: Do you check any difference among groups in gender and age? CHD group in Table 1 seems to have higher median age compared to DS groups (both CHD+ and CHD-).

Response 2: because this was a study of birth defects, we did not examine the difference between groups in age. We examined the difference between the groups (DSCHD+ and CHD) by sex; no difference in sex distribution was found, and the data were not shown in the manuscript. The CHD group had a higher mean age compared with the groups DS (both CHD+ and CHD-), because the CHD group had a smaller number of subjects at birth compared with the groups DS.

Point 3: Reference 3. The first part of the title is missing.

Response 3:  3.  Liu, Y.; Chen, S.; Zühlke, L.; Black, G. C.; Choy, M. K.; Li, N.; Keavney, B. D. Global birth prevalence of congenital heart defects 1970-2017: updated systematic review and meta-analysis of 260 studies. International journal of epidemiology 2019, 48, 455–463. https://doi.org/10.1093/ije/dyz009 – A sentence (Global) has been added at the beginning of the article title

Point 4: The table numbers in the text are wrong: line 166, you have to correct "Table 1" with "Table 3" and line 171 "Table 2" with "Table 4".

Response 4: we are sorry, but we do not understand what is wrong in line 166 and 171. Could you please give us more specific instructions on what we need to change.

Point 5: Line 144-145. “Analyzed” is reported twice.

Response 5:  Suggestion accepted. This sentence has been deleted.

Response to Reviewer 1 Comments

We would like to thank the reviewer for valuable comments and observations, which improved our paper. We appreciate the time and effort spent in analyzing our paper. Please find detailed response below. All changes in the manuscript are marked up using the “Track Changes” function.

Point 1: I suggest the authors to add a schematic figure of folate/methionine/homocysteine ​​metabolism with related enzymes to better understand the metabolic pathway which is well explained in the introductory section.

Response 1: we add a schematic figure of folate/methionine/homocysteine metabolism with related enzymes, as recommended.

Point 2: Do you check any difference among groups in gender and age? CHD group in Table 1 seems to have higher median age compared to DS groups (both CHD+ and CHD-).

Response 2: because this was a study of birth defects, we did not examine the difference between groups in age. We examined the difference between the groups (DSCHD+ and CHD) by sex; no difference in sex distribution was found, and the data were not shown in the manuscript. The CHD group had a higher mean age compared with the groups DS (both CHD+ and CHD-), because the CHD group had a smaller number of subjects at birth compared with the groups DS.

Point 3: Reference 3. The first part of the title is missing.

Response 3:  3.  Liu, Y.; Chen, S.; Zühlke, L.; Black, G. C.; Choy, M. K.; Li, N.; Keavney, B. D. Global birth prevalence of congenital heart defects 1970-2017: updated systematic review and meta-analysis of 260 studies. International journal of epidemiology 2019, 48, 455–463. https://doi.org/10.1093/ije/dyz009 – A sentence (Global) has been added at the beginning of the article title

Point 4: The table numbers in the text are wrong: line 166, you have to correct "Table 1" with "Table 3" and line 171 "Table 2" with "Table 4".

Response 4: we are sorry, but we do not understand what is wrong in line 166 and 171. Could you please give us more specific instructions on what we need to change.

Point 5: Line 144-145. “Analyzed” is reported twice.

Response 5:  Suggestion accepted. This sentence has been deleted.

Response to Reviewer 1 Comments

We would like to thank the reviewer for valuable comments and observations, which improved our paper. We appreciate the time and effort spent in analyzing our paper. Please find detailed response below. All changes in the manuscript are marked up using the “Track Changes” function.

Point 1: I suggest the authors to add a schematic figure of folate/methionine/homocysteine ​​metabolism with related enzymes to better understand the metabolic pathway which is well explained in the introductory section.

Response 1: we add a schematic figure of folate/methionine/homocysteine metabolism with related enzymes, as recommended.

Point 2: Do you check any difference among groups in gender and age? CHD group in Table 1 seems to have higher median age compared to DS groups (both CHD+ and CHD-).

Response 2: because this was a study of birth defects, we did not examine the difference between groups in age. We examined the difference between the groups (DSCHD+ and CHD) by sex; no difference in sex distribution was found, and the data were not shown in the manuscript. The CHD group had a higher mean age compared with the groups DS (both CHD+ and CHD-), because the CHD group had a smaller number of subjects at birth compared with the groups DS.

Point 3: Reference 3. The first part of the title is missing.

Response 3:  3.  Liu, Y.; Chen, S.; Zühlke, L.; Black, G. C.; Choy, M. K.; Li, N.; Keavney, B. D. Global birth prevalence of congenital heart defects 1970-2017: updated systematic review and meta-analysis of 260 studies. International journal of epidemiology 2019, 48, 455–463. https://doi.org/10.1093/ije/dyz009 – A sentence (Global) has been added at the beginning of the article title

Point 4: The table numbers in the text are wrong: line 166, you have to correct "Table 1" with "Table 3" and line 171 "Table 2" with "Table 4".

Response 4: we are sorry, but we do not understand what is wrong in line 166 and 171. Could you please give us more specific instructions on what we need to change.

Point 5: Line 144-145. “Analyzed” is reported twice.

Response 5:  Suggestion accepted. This sentence has been deleted.

Response to Reviewer 1 Comments

We would like to thank the reviewer for valuable comments and observations, which improved our paper. We appreciate the time and effort spent in analyzing our paper. Please find detailed response below. All changes in the manuscript are marked up using the “Track Changes” function.

Point 1: I suggest the authors to add a schematic figure of folate/methionine/homocysteine ​​metabolism with related enzymes to better understand the metabolic pathway which is well explained in the introductory section.

Response 1: we add a schematic figure of folate/methionine/homocysteine metabolism with related enzymes, as recommended.

Point 2: Do you check any difference among groups in gender and age? CHD group in Table 1 seems to have higher median age compared to DS groups (both CHD+ and CHD-).

Response 2: because this was a study of birth defects, we did not examine the difference between groups in age. We examined the difference between the groups (DSCHD+ and CHD) by sex; no difference in sex distribution was found, and the data were not shown in the manuscript. The CHD group had a higher mean age compared with the groups DS (both CHD+ and CHD-), because the CHD group had a smaller number of subjects at birth compared with the groups DS.

Point 3: Reference 3. The first part of the title is missing.

Response 3:  3.  Liu, Y.; Chen, S.; Zühlke, L.; Black, G. C.; Choy, M. K.; Li, N.; Keavney, B. D. Global birth prevalence of congenital heart defects 1970-2017: updated systematic review and meta-analysis of 260 studies. International journal of epidemiology 2019, 48, 455–463. https://doi.org/10.1093/ije/dyz009 – A sentence (Global) has been added at the beginning of the article title

Point 4: The table numbers in the text are wrong: line 166, you have to correct "Table 1" with "Table 3" and line 171 "Table 2" with "Table 4".

Response 4: we are sorry, but we do not understand what is wrong in line 166 and 171. Could you please give us more specific instructions on what we need to change.

Point 5: Line 144-145. “Analyzed” is reported twice.

Response 5:  Suggestion accepted. This sentence has been deleted.

Response to Reviewer 1 Comments

We would like to thank the reviewer for valuable comments and observations, which improved our paper. We appreciate the time and effort spent in analyzing our paper. Please find detailed response below. All changes in the manuscript are marked up using the “Track Changes” function.

Point 1: I suggest the authors to add a schematic figure of folate/methionine/homocysteine ​​metabolism with related enzymes to better understand the metabolic pathway which is well explained in the introductory section.

Response 1: we add a schematic figure of folate/methionine/homocysteine metabolism with related enzymes, as recommended.

Point 2: Do you check any difference among groups in gender and age? CHD group in Table 1 seems to have higher median age compared to DS groups (both CHD+ and CHD-).

Response 2: because this was a study of birth defects, we did not examine the difference between groups in age. We examined the difference between the groups (DSCHD+ and CHD) by sex; no difference in sex distribution was found, and the data were not shown in the manuscript. The CHD group had a higher mean age compared with the groups DS (both CHD+ and CHD-), because the CHD group had a smaller number of subjects at birth compared with the groups DS.

Point 3: Reference 3. The first part of the title is missing.

Response 3:  3.  Liu, Y.; Chen, S.; Zühlke, L.; Black, G. C.; Choy, M. K.; Li, N.; Keavney, B. D. Global birth prevalence of congenital heart defects 1970-2017: updated systematic review and meta-analysis of 260 studies. International journal of epidemiology 2019, 48, 455–463. https://doi.org/10.1093/ije/dyz009 – A sentence (Global) has been added at the beginning of the article title

Point 4: The table numbers in the text are wrong: line 166, you have to correct "Table 1" with "Table 3" and line 171 "Table 2" with "Table 4".

Response 4: we are sorry, but we do not understand what is wrong in line 166 and 171. Could you please give us more specific instructions on what we need to change.

Point 5: Line 144-145. “Analyzed” is reported twice.

Response 5:  Suggestion accepted. This sentence has been deleted.

Response to Reviewer 1 Comments

We would like to thank the reviewer for valuable comments and observations, which improved our paper. We appreciate the time and effort spent in analyzing our paper. Please find detailed response below. All changes in the manuscript are marked up using the “Track Changes” function.

Point 1: I suggest the authors to add a schematic figure of folate/methionine/homocysteine ​​metabolism with related enzymes to better understand the metabolic pathway which is well explained in the introductory section.

Response 1: we add a schematic figure of folate/methionine/homocysteine metabolism with related enzymes, as recommended.

Point 2: Do you check any difference among groups in gender and age? CHD group in Table 1 seems to have higher median age compared to DS groups (both CHD+ and CHD-).

Response 2: because this was a study of birth defects, we did not examine the difference between groups in age. We examined the difference between the groups (DSCHD+ and CHD) by sex; no difference in sex distribution was found, and the data were not shown in the manuscript. The CHD group had a higher mean age compared with the groups DS (both CHD+ and CHD-), because the CHD group had a smaller number of subjects at birth compared with the groups DS.

Point 3: Reference 3. The first part of the title is missing.

Response 3:  3.  Liu, Y.; Chen, S.; Zühlke, L.; Black, G. C.; Choy, M. K.; Li, N.; Keavney, B. D. Global birth prevalence of congenital heart defects 1970-2017: updated systematic review and meta-analysis of 260 studies. International journal of epidemiology 2019, 48, 455–463. https://doi.org/10.1093/ije/dyz009 – A sentence (Global) has been added at the beginning of the article title

Point 4: The table numbers in the text are wrong: line 166, you have to correct "Table 1" with "Table 3" and line 171 "Table 2" with "Table 4".

Response 4: we are sorry, but we do not understand what is wrong in line 166 and 171. Could you please give us more specific instructions on what we need to change.

Point 5: Line 144-145. “Analyzed” is reported twice.

Response 5:  Suggestion accepted. This sentence has been deleted.

Response to Reviewer 1 Comments

We would like to thank the reviewer for valuable comments and observations, which improved our paper. We appreciate the time and effort spent in analyzing our paper. Please find detailed response below. All changes in the manuscript are marked up using the “Track Changes” function.

Point 1: I suggest the authors to add a schematic figure of folate/methionine/homocysteine ​​metabolism with related enzymes to better understand the metabolic pathway which is well explained in the introductory section.

Response 1: we add a schematic figure of folate/methionine/homocysteine metabolism with related enzymes, as recommended.

Point 2: Do you check any difference among groups in gender and age? CHD group in Table 1 seems to have higher median age compared to DS groups (both CHD+ and CHD-).

Response 2: because this was a study of birth defects, we did not examine the difference between groups in age. We examined the difference between the groups (DSCHD+ and CHD) by sex; no difference in sex distribution was found, and the data were not shown in the manuscript. The CHD group had a higher mean age compared with the groups DS (both CHD+ and CHD-), because the CHD group had a smaller number of subjects at birth compared with the groups DS.

Point 3: Reference 3. The first part of the title is missing.

Response 3:  3.  Liu, Y.; Chen, S.; Zühlke, L.; Black, G. C.; Choy, M. K.; Li, N.; Keavney, B. D. Global birth prevalence of congenital heart defects 1970-2017: updated systematic review and meta-analysis of 260 studies. International journal of epidemiology 2019, 48, 455–463. https://doi.org/10.1093/ije/dyz009 – A sentence (Global) has been added at the beginning of the article title

Point 4: The table numbers in the text are wrong: line 166, you have to correct "Table 1" with "Table 3" and line 171 "Table 2" with "Table 4".

Response 4: we are sorry, but we do not understand what is wrong in line 166 and 171. Could you please give us more specific instructions on what we need to change.

Point 5: Line 144-145. “Analyzed” is reported twice.

Response 5:  Suggestion accepted. This sentence has been deleted.

Response to Reviewer 1 Comments

We would like to thank the reviewer for valuable comments and observations, which improved our paper. We appreciate the time and effort spent in analyzing our paper. Please find detailed response below. All changes in the manuscript are marked up using the “Track Changes” function.

Point 1: I suggest the authors to add a schematic figure of folate/methionine/homocysteine ​​metabolism with related enzymes to better understand the metabolic pathway which is well explained in the introductory section.

Response 1: we add a schematic figure of folate/methionine/homocysteine metabolism with related enzymes, as recommended.

Point 2: Do you check any difference among groups in gender and age? CHD group in Table 1 seems to have higher median age compared to DS groups (both CHD+ and CHD-).

Response 2: because this was a study of birth defects, we did not examine the difference between groups in age. We examined the difference between the groups (DSCHD+ and CHD) by sex; no difference in sex distribution was found, and the data were not shown in the manuscript. The CHD group had a higher mean age compared with the groups DS (both CHD+ and CHD-), because the CHD group had a smaller number of subjects at birth compared with the groups DS.

Point 3: Reference 3. The first part of the title is missing.

Response 3:  3.  Liu, Y.; Chen, S.; Zühlke, L.; Black, G. C.; Choy, M. K.; Li, N.; Keavney, B. D. Global birth prevalence of congenital heart defects 1970-2017: updated systematic review and meta-analysis of 260 studies. International journal of epidemiology 2019, 48, 455–463. https://doi.org/10.1093/ije/dyz009 – A sentence (Global) has been added at the beginning of the article title

Point 4: The table numbers in the text are wrong: line 166, you have to correct "Table 1" with "Table 3" and line 171 "Table 2" with "Table 4".

Response 4: we are sorry, but we do not understand what is wrong in line 166 and 171. Could you please give us more specific instructions on what we need to change.

Point 5: Line 144-145. “Analyzed” is reported twice.

Response 5:  Suggestion accepted. This sentence has been deleted.

Response to Reviewer 1 Comments

We would like to thank the reviewer for valuable comments and observations, which improved our paper. We appreciate the time and effort spent in analyzing our paper. Please find detailed response below. All changes in the manuscript are marked up using the “Track Changes” function.

Point 1: I suggest the authors to add a schematic figure of folate/methionine/homocysteine ​​metabolism with related enzymes to better understand the metabolic pathway which is well explained in the introductory section.

Response 1: we add a schematic figure of folate/methionine/homocysteine metabolism with related enzymes, as recommended.

Point 2: Do you check any difference among groups in gender and age? CHD group in Table 1 seems to have higher median age compared to DS groups (both CHD+ and CHD-).

Response 2: because this was a study of birth defects, we did not examine the difference between groups in age. We examined the difference between the groups (DSCHD+ and CHD) by sex; no difference in sex distribution was found, and the data were not shown in the manuscript. The CHD group had a higher mean age compared with the groups DS (both CHD+ and CHD-), because the CHD group had a smaller number of subjects at birth compared with the groups DS.

Point 3: Reference 3. The first part of the title is missing.

Response 3:  3.  Liu, Y.; Chen, S.; Zühlke, L.; Black, G. C.; Choy, M. K.; Li, N.; Keavney, B. D. Global birth prevalence of congenital heart defects 1970-2017: updated systematic review and meta-analysis of 260 studies. International journal of epidemiology 2019, 48, 455–463. https://doi.org/10.1093/ije/dyz009 – A sentence (Global) has been added at the beginning of the article title

Point 4: The table numbers in the text are wrong: line 166, you have to correct "Table 1" with "Table 3" and line 171 "Table 2" with "Table 4".

Response 4: we are sorry, but we do not understand what is wrong in line 166 and 171. Could you please give us more specific instructions on what we need to change.

Point 5: Line 144-145. “Analyzed” is reported twice.

Response 5:  Suggestion accepted. This sentence has been deleted.

Response to Reviewer 1 Comments

We would like to thank the reviewer for valuable comments and observations, which improved our paper. We appreciate the time and effort spent in analyzing our paper. Please find detailed response below. All changes in the manuscript are marked up using the “Track Changes” function.

Point 1: I suggest the authors to add a schematic figure of folate/methionine/homocysteine ​​metabolism with related enzymes to better understand the metabolic pathway which is well explained in the introductory section.

Response 1: we add a schematic figure of folate/methionine/homocysteine metabolism with related enzymes, as recommended.

Point 2: Do you check any difference among groups in gender and age? CHD group in Table 1 seems to have higher median age compared to DS groups (both CHD+ and CHD-).

Response 2: because this was a study of birth defects, we did not examine the difference between groups in age. We examined the difference between the groups (DSCHD+ and CHD) by sex; no difference in sex distribution was found, and the data were not shown in the manuscript. The CHD group had a higher mean age compared with the groups DS (both CHD+ and CHD-), because the CHD group had a smaller number of subjects at birth compared with the groups DS.

Point 3: Reference 3. The first part of the title is missing.

Response 3:  3.  Liu, Y.; Chen, S.; Zühlke, L.; Black, G. C.; Choy, M. K.; Li, N.; Keavney, B. D. Global birth prevalence of congenital heart defects 1970-2017: updated systematic review and meta-analysis of 260 studies. International journal of epidemiology 2019, 48, 455–463. https://doi.org/10.1093/ije/dyz009 – A sentence (Global) has been added at the beginning of the article title

Point 4: The table numbers in the text are wrong: line 166, you have to correct "Table 1" with "Table 3" and line 171 "Table 2" with "Table 4".

Response 4: we are sorry, but we do not understand what is wrong in line 166 and 171. Could you please give us more specific instructions on what we need to change.

Point 5: Line 144-145. “Analyzed” is reported twice.

Response 5:  Suggestion accepted. This sentence has been deleted.

Response to Reviewer 1 Comments

We would like to thank the reviewer for valuable comments and observations, which improved our paper. We appreciate the time and effort spent in analyzing our paper. Please find detailed response below. All changes in the manuscript are marked up using the “Track Changes” function.

Point 1: I suggest the authors to add a schematic figure of folate/methionine/homocysteine ​​metabolism with related enzymes to better understand the metabolic pathway which is well explained in the introductory section.

Response 1: we add a schematic figure of folate/methionine/homocysteine metabolism with related enzymes, as recommended.

Point 2: Do you check any difference among groups in gender and age? CHD group in Table 1 seems to have higher median age compared to DS groups (both CHD+ and CHD-).

Response 2: because this was a study of birth defects, we did not examine the difference between groups in age. We examined the difference between the groups (DSCHD+ and CHD) by sex; no difference in sex distribution was found, and the data were not shown in the manuscript. The CHD group had a higher mean age compared with the groups DS (both CHD+ and CHD-), because the CHD group had a smaller number of subjects at birth compared with the groups DS.

Point 3: Reference 3. The first part of the title is missing.

Response 3:  3.  Liu, Y.; Chen, S.; Zühlke, L.; Black, G. C.; Choy, M. K.; Li, N.; Keavney, B. D. Global birth prevalence of congenital heart defects 1970-2017: updated systematic review and meta-analysis of 260 studies. International journal of epidemiology 2019, 48, 455–463. https://doi.org/10.1093/ije/dyz009 – A sentence (Global) has been added at the beginning of the article title

Point 4: The table numbers in the text are wrong: line 166, you have to correct "Table 1" with "Table 3" and line 171 "Table 2" with "Table 4".

Response 4: we are sorry, but we do not understand what is wrong in line 166 and 171. Could you please give us more specific instructions on what we need to change.

Point 5: Line 144-145. “Analyzed” is reported twice.

Response 5:  Suggestion accepted. This sentence has been deleted.

Response to Reviewer 1 Comments

We would like to thank the reviewer for valuable comments and observations, which improved our paper. We appreciate the time and effort spent in analyzing our paper. Please find detailed response below. All changes in the manuscript are marked up using the “Track Changes” function.

Point 1: I suggest the authors to add a schematic figure of folate/methionine/homocysteine ​​metabolism with related enzymes to better understand the metabolic pathway which is well explained in the introductory section.

Response 1: we add a schematic figure of folate/methionine/homocysteine metabolism with related enzymes, as recommended.

Point 2: Do you check any difference among groups in gender and age? CHD group in Table 1 seems to have higher median age compared to DS groups (both CHD+ and CHD-).

Response 2: because this was a study of birth defects, we did not examine the difference between groups in age. We examined the difference between the groups (DSCHD+ and CHD) by sex; no difference in sex distribution was found, and the data were not shown in the manuscript. The CHD group had a higher mean age compared with the groups DS (both CHD+ and CHD-), because the CHD group had a smaller number of subjects at birth compared with the groups DS.

Point 3: Reference 3. The first part of the title is missing.

Response 3:  3.  Liu, Y.; Chen, S.; Zühlke, L.; Black, G. C.; Choy, M. K.; Li, N.; Keavney, B. D. Global birth prevalence of congenital heart defects 1970-2017: updated systematic review and meta-analysis of 260 studies. International journal of epidemiology 2019, 48, 455–463. https://doi.org/10.1093/ije/dyz009 – A sentence (Global) has been added at the beginning of the article title

Point 4: The table numbers in the text are wrong: line 166, you have to correct "Table 1" with "Table 3" and line 171 "Table 2" with "Table 4".

Response 4: we are sorry, but we do not understand what is wrong in line 166 and 171. Could you please give us more specific instructions on what we need to change.

Point 5: Line 144-145. “Analyzed” is reported twice.

Response 5:  Suggestion accepted. This sentence has been deleted.

Response to Reviewer 1 Comments

We would like to thank the reviewer for valuable comments and observations, which improved our paper. We appreciate the time and effort spent in analyzing our paper. Please find detailed response below. All changes in the manuscript are marked up using the “Track Changes” function.

Point 1: I suggest the authors to add a schematic figure of folate/methionine/homocysteine ​​metabolism with related enzymes to better understand the metabolic pathway which is well explained in the introductory section.

Response 1: we add a schematic figure of folate/methionine/homocysteine metabolism with related enzymes, as recommended.

Point 2: Do you check any difference among groups in gender and age? CHD group in Table 1 seems to have higher median age compared to DS groups (both CHD+ and CHD-).

Response 2: because this was a study of birth defects, we did not examine the difference between groups in age. We examined the difference between the groups (DSCHD+ and CHD) by sex; no difference in sex distribution was found, and the data were not shown in the manuscript. The CHD group had a higher mean age compared with the groups DS (both CHD+ and CHD-), because the CHD group had a smaller number of subjects at birth compared with the groups DS.

Point 3: Reference 3. The first part of the title is missing.

Response 3:  3.  Liu, Y.; Chen, S.; Zühlke, L.; Black, G. C.; Choy, M. K.; Li, N.; Keavney, B. D. Global birth prevalence of congenital heart defects 1970-2017: updated systematic review and meta-analysis of 260 studies. International journal of epidemiology 2019, 48, 455–463. https://doi.org/10.1093/ije/dyz009 – A sentence (Global) has been added at the beginning of the article title

Point 4: The table numbers in the text are wrong: line 166, you have to correct "Table 1" with "Table 3" and line 171 "Table 2" with "Table 4".

Response 4: we are sorry, but we do not understand what is wrong in line 166 and 171. Could you please give us more specific instructions on what we need to change.

Point 5: Line 144-145. “Analyzed” is reported twice.

Response 5:  Suggestion accepted. This sentence has been deleted.

Response to Reviewer 1 Comments

We would like to thank the reviewer for valuable comments and observations, which improved our paper. We appreciate the time and effort spent in analyzing our paper. Please find detailed response below. All changes in the manuscript are marked up using the “Track Changes” function.

Point 1: I suggest the authors to add a schematic figure of folate/methionine/homocysteine ​​metabolism with related enzymes to better understand the metabolic pathway which is well explained in the introductory section.

Response 1: we add a schematic figure of folate/methionine/homocysteine metabolism with related enzymes, as recommended.

Point 2: Do you check any difference among groups in gender and age? CHD group in Table 1 seems to have higher median age compared to DS groups (both CHD+ and CHD-).

Response 2: because this was a study of birth defects, we did not examine the difference between groups in age. We examined the difference between the groups (DSCHD+ and CHD) by sex; no difference in sex distribution was found, and the data were not shown in the manuscript. The CHD group had a higher mean age compared with the groups DS (both CHD+ and CHD-), because the CHD group had a smaller number of subjects at birth compared with the groups DS.

Point 3: Reference 3. The first part of the title is missing.

Response 3:  3.  Liu, Y.; Chen, S.; Zühlke, L.; Black, G. C.; Choy, M. K.; Li, N.; Keavney, B. D. Global birth prevalence of congenital heart defects 1970-2017: updated systematic review and meta-analysis of 260 studies. International journal of epidemiology 2019, 48, 455–463. https://doi.org/10.1093/ije/dyz009 – A sentence (Global) has been added at the beginning of the article title

Point 4: The table numbers in the text are wrong: line 166, you have to correct "Table 1" with "Table 3" and line 171 "Table 2" with "Table 4".

Response 4: we are sorry, but we do not understand what is wrong in line 166 and 171. Could you please give us more specific instructions on what we need to change.

Point 5: Line 144-145. “Analyzed” is reported twice.

Response 5:  Suggestion accepted. This sentence has been deleted.

Response to Reviewer 1 Comments

We would like to thank the reviewer for valuable comments and observations, which improved our paper. We appreciate the time and effort spent in analyzing our paper. Please find detailed response below. All changes in the manuscript are marked up using the “Track Changes” function.

Point 1: I suggest the authors to add a schematic figure of folate/methionine/homocysteine ​​metabolism with related enzymes to better understand the metabolic pathway which is well explained in the introductory section.

Response 1: we add a schematic figure of folate/methionine/homocysteine metabolism with related enzymes, as recommended.

Point 2: Do you check any difference among groups in gender and age? CHD group in Table 1 seems to have higher median age compared to DS groups (both CHD+ and CHD-).

Response 2: because this was a study of birth defects, we did not examine the difference between groups in age. We examined the difference between the groups (DSCHD+ and CHD) by sex; no difference in sex distribution was found, and the data were not shown in the manuscript. The CHD group had a higher mean age compared with the groups DS (both CHD+ and CHD-), because the CHD group had a smaller number of subjects at birth compared with the groups DS.

Point 3: Reference 3. The first part of the title is missing.

Response 3:  3.  Liu, Y.; Chen, S.; Zühlke, L.; Black, G. C.; Choy, M. K.; Li, N.; Keavney, B. D. Global birth prevalence of congenital heart defects 1970-2017: updated systematic review and meta-analysis of 260 studies. International journal of epidemiology 2019, 48, 455–463. https://doi.org/10.1093/ije/dyz009 – A sentence (Global) has been added at the beginning of the article title

Point 4: The table numbers in the text are wrong: line 166, you have to correct "Table 1" with "Table 3" and line 171 "Table 2" with "Table 4".

Response 4: we are sorry, but we do not understand what is wrong in line 166 and 171. Could you please give us more specific instructions on what we need to change.

Point 5: Line 144-145. “Analyzed” is reported twice.

Response 5:  Suggestion accepted. This sentence has been deleted.

Response to Reviewer 1 Comments

We would like to thank the reviewer for valuable comments and observations, which improved our paper. We appreciate the time and effort spent in analyzing our paper. Please find detailed response below. All changes in the manuscript are marked up using the “Track Changes” function.

Point 1: I suggest the authors to add a schematic figure of folate/methionine/homocysteine ​​metabolism with related enzymes to better understand the metabolic pathway which is well explained in the introductory section.

Response 1: we add a schematic figure of folate/methionine/homocysteine metabolism with related enzymes, as recommended.

Point 2: Do you check any difference among groups in gender and age? CHD group in Table 1 seems to have higher median age compared to DS groups (both CHD+ and CHD-).

Response 2: because this was a study of birth defects, we did not examine the difference between groups in age. We examined the difference between the groups (DSCHD+ and CHD) by sex; no difference in sex distribution was found, and the data were not shown in the manuscript. The CHD group had a higher mean age compared with the groups DS (both CHD+ and CHD-), because the CHD group had a smaller number of subjects at birth compared with the groups DS.

Point 3: Reference 3. The first part of the title is missing.

Response 3:  3.  Liu, Y.; Chen, S.; Zühlke, L.; Black, G. C.; Choy, M. K.; Li, N.; Keavney, B. D. Global birth prevalence of congenital heart defects 1970-2017: updated systematic review and meta-analysis of 260 studies. International journal of epidemiology 2019, 48, 455–463. https://doi.org/10.1093/ije/dyz009 – A sentence (Global) has been added at the beginning of the article title

Point 4: The table numbers in the text are wrong: line 166, you have to correct "Table 1" with "Table 3" and line 171 "Table 2" with "Table 4".

Response 4: we are sorry, but we do not understand what is wrong in line 166 and 171. Could you please give us more specific instructions on what we need to change.

Point 5: Line 144-145. “Analyzed” is reported twice.

Response 5:  Suggestion accepted. This sentence has been deleted.

Response to Reviewer 1 Comments

We would like to thank the reviewer for valuable comments and observations, which improved our paper. We appreciate the time and effort spent in analyzing our paper. Please find detailed response below. All changes in the manuscript are marked up using the “Track Changes” function.

Point 1: I suggest the authors to add a schematic figure of folate/methionine/homocysteine ​​metabolism with related enzymes to better understand the metabolic pathway which is well explained in the introductory section.

Response 1: we add a schematic figure of folate/methionine/homocysteine metabolism with related enzymes, as recommended.

Point 2: Do you check any difference among groups in gender and age? CHD group in Table 1 seems to have higher median age compared to DS groups (both CHD+ and CHD-).

Response 2: because this was a study of birth defects, we did not examine the difference between groups in age. We examined the difference between the groups (DSCHD+ and CHD) by sex; no difference in sex distribution was found, and the data were not shown in the manuscript. The CHD group had a higher mean age compared with the groups DS (both CHD+ and CHD-), because the CHD group had a smaller number of subjects at birth compared with the groups DS.

Point 3: Reference 3. The first part of the title is missing.

Response 3:  3.  Liu, Y.; Chen, S.; Zühlke, L.; Black, G. C.; Choy, M. K.; Li, N.; Keavney, B. D. Global birth prevalence of congenital heart defects 1970-2017: updated systematic review and meta-analysis of 260 studies. International journal of epidemiology 2019, 48, 455–463. https://doi.org/10.1093/ije/dyz009 – A sentence (Global) has been added at the beginning of the article title

Point 4: The table numbers in the text are wrong: line 166, you have to correct "Table 1" with "Table 3" and line 171 "Table 2" with "Table 4".

Response 4: we are sorry, but we do not understand what is wrong in line 166 and 171. Could you please give us more specific instructions on what we need to change.

Point 5: Line 144-145. “Analyzed” is reported twice.

Response 5:  Suggestion accepted. This sentence has been deleted.

Response to Reviewer 1 Comments

We would like to thank the reviewer for valuable comments and observations, which improved our paper. We appreciate the time and effort spent in analyzing our paper. Please find detailed response below. All changes in the manuscript are marked up using the “Track Changes” function.

Point 1: I suggest the authors to add a schematic figure of folate/methionine/homocysteine ​​metabolism with related enzymes to better understand the metabolic pathway which is well explained in the introductory section.

Response 1: we add a schematic figure of folate/methionine/homocysteine metabolism with related enzymes, as recommended.

Point 2: Do you check any difference among groups in gender and age? CHD group in Table 1 seems to have higher median age compared to DS groups (both CHD+ and CHD-).

Response 2: because this was a study of birth defects, we did not examine the difference between groups in age. We examined the difference between the groups (DSCHD+ and CHD) by sex; no difference in sex distribution was found, and the data were not shown in the manuscript. The CHD group had a higher mean age compared with the groups DS (both CHD+ and CHD-), because the CHD group had a smaller number of subjects at birth compared with the groups DS.

Point 3: Reference 3. The first part of the title is missing.

Response 3:  3.  Liu, Y.; Chen, S.; Zühlke, L.; Black, G. C.; Choy, M. K.; Li, N.; Keavney, B. D. Global birth prevalence of congenital heart defects 1970-2017: updated systematic review and meta-analysis of 260 studies. International journal of epidemiology 2019, 48, 455–463. https://doi.org/10.1093/ije/dyz009 – A sentence (Global) has been added at the beginning of the article title

Point 4: The table numbers in the text are wrong: line 166, you have to correct "Table 1" with "Table 3" and line 171 "Table 2" with "Table 4".

Response 4: we are sorry, but we do not understand what is wrong in line 166 and 171. Could you please give us more specific instructions on what we need to change.

Point 5: Line 144-145. “Analyzed” is reported twice.

Response 5:  Suggestion accepted. This sentence has been deleted.

Response to Reviewer 1 Comments

We would like to thank the reviewer for valuable comments and observations, which improved our paper. We appreciate the time and effort spent in analyzing our paper. Please find detailed response below. All changes in the manuscript are marked up using the “Track Changes” function.

Point 1: I suggest the authors to add a schematic figure of folate/methionine/homocysteine ​​metabolism with related enzymes to better understand the metabolic pathway which is well explained in the introductory section.

Response 1: we add a schematic figure of folate/methionine/homocysteine metabolism with related enzymes, as recommended.

Point 2: Do you check any difference among groups in gender and age? CHD group in Table 1 seems to have higher median age compared to DS groups (both CHD+ and CHD-).

Response 2: because this was a study of birth defects, we did not examine the difference between groups in age. We examined the difference between the groups (DSCHD+ and CHD) by sex; no difference in sex distribution was found, and the data were not shown in the manuscript. The CHD group had a higher mean age compared with the groups DS (both CHD+ and CHD-), because the CHD group had a smaller number of subjects at birth compared with the groups DS.

Point 3: Reference 3. The first part of the title is missing.

Response 3:  3.  Liu, Y.; Chen, S.; Zühlke, L.; Black, G. C.; Choy, M. K.; Li, N.; Keavney, B. D. Global birth prevalence of congenital heart defects 1970-2017: updated systematic review and meta-analysis of 260 studies. International journal of epidemiology 2019, 48, 455–463. https://doi.org/10.1093/ije/dyz009 – A sentence (Global) has been added at the beginning of the article title

Point 4: The table numbers in the text are wrong: line 166, you have to correct "Table 1" with "Table 3" and line 171 "Table 2" with "Table 4".

Response 4: we are sorry, but we do not understand what is wrong in line 166 and 171. Could you please give us more specific instructions on what we need to change.

Point 5: Line 144-145. “Analyzed” is reported twice.

Response 5:  Suggestion accepted. This sentence has been deleted.

Response to Reviewer 1 Comments

We would like to thank the reviewer for valuable comments and observations, which improved our paper. We appreciate the time and effort spent in analyzing our paper. Please find detailed response below. All changes in the manuscript are marked up using the “Track Changes” function.

Point 1: I suggest the authors to add a schematic figure of folate/methionine/homocysteine ​​metabolism with related enzymes to better understand the metabolic pathway which is well explained in the introductory section.

Response 1: we add a schematic figure of folate/methionine/homocysteine metabolism with related enzymes, as recommended.

Point 2: Do you check any difference among groups in gender and age? CHD group in Table 1 seems to have higher median age compared to DS groups (both CHD+ and CHD-).

Response 2: because this was a study of birth defects, we did not examine the difference between groups in age. We examined the difference between the groups (DSCHD+ and CHD) by sex; no difference in sex distribution was found, and the data were not shown in the manuscript. The CHD group had a higher mean age compared with the groups DS (both CHD+ and CHD-), because the CHD group had a smaller number of subjects at birth compared with the groups DS.

Point 3: Reference 3. The first part of the title is missing.

Response 3:  3.  Liu, Y.; Chen, S.; Zühlke, L.; Black, G. C.; Choy, M. K.; Li, N.; Keavney, B. D. Global birth prevalence of congenital heart defects 1970-2017: updated systematic review and meta-analysis of 260 studies. International journal of epidemiology 2019, 48, 455–463. https://doi.org/10.1093/ije/dyz009 – A sentence (Global) has been added at the beginning of the article title

Point 4: The table numbers in the text are wrong: line 166, you have to correct "Table 1" with "Table 3" and line 171 "Table 2" with "Table 4".

Response 4: we are sorry, but we do not understand what is wrong in line 166 and 171. Could you please give us more specific instructions on what we need to change.

Point 5: Line 144-145. “Analyzed” is reported twice.

Response 5:  Suggestion accepted. This sentence has been deleted.

Reviewer 2 Report

The paper is interesting because it concerns a topic of great clinical interest: i.e.Hthe presence of CDH in DS.

However, some revisions are needed.

In the introduction the authors must justify the choice of polymorphisms: in other words, they must justify why they specifically target the 7 polymorphisms

Sample size is missing.

Discussion needs to be more cautious. The conclusions must be softened because the sample size is low

Table 1

the percentages in columns DSCHD+ and DSCHD- are incorrect

Author Response

Response to Reviewer 2 Comments

We would like to thank the reviewer for valuable comments and observations, which improved our paper. We appreciate the time and effort spent in analyzing our paper. Please find detailed response below. All changes in the manuscript are marked up using the “Track Changes” function.

Point 1: In the introduction the authors must justify the choice of polymorphisms: in other words, they must justify why they specifically target the 7 polymorphisms

Response 1: we selected SNPs considering their influence on methylation during development and later in life, in order to justify why we specifically select these polymorphisms, we expanded the introduction with a description of the role of the selected SNPs in folate and homocysteine metabolism, the consequent influence on methylation and the development of CHD.

Point 2: Sample size is missing

Response 2: the sample size is specified in section 2. Materials and Methods 2.1. Patients

This study was conducted on 350 participants, including 134 DS individuals with CHD (DSCHD+) and 124 DS individuals without CHD (DSCHD-), and 92 individuals with non-syndromic CHD (Table 1). In the section Results, the number of cases for each group is listed in Table 1. Characteristics of study populations

Point 3: Discussion needs to be more cautious. The conclusions must be softened because the sample size is low

Response 3:

We fully agree with the reviewer that this statement was too strong.

Now we rephrased the sentence as “Furthermore, our results suggest that the DNMT3B rs1569686 TT genotype may increases CHD risk in individuals with non-syndromic CHD compared to the DS group with CHD, which was also found under the dominant and codominant genetic models of the same polymorphism”.

We softened the sentence in the conclusion in a way “Due to the fact that research is carried out on a small sample size, further studies in larger cohorts focused on the specific types of CHDs are needed to define functional roles of MTHFR, MTRR, and DNMT genes polymorphisms in CHD etiology

Point 4: Table 1 the percentages in columns DSCHD+ and DSCHD- are incorrect

Response 4: We changed percentages.

Reviewer 3 Report

The study  entitled "DNMT3B rs2424913 as a Risk Factor for Congenital Heart  Defects in Down Syndrome" is based on correlation assessment between MTHFR, MTRR, and DNMT gene polymorphisms and CHD in Down syndrome. The research was conducted according to ethically approved protocols. 124 DS  individuals were recruited without CHD (DSCHD-), and 92 individuals with non-syndromic CHD with their informed consents. 

their results concluded that  DNMT3B rs2424913 TT genotypes and the genetic model CC + CT vs TT could be a  predisposing factor for CHD, in individuals with Down syndrome, particularly those with  ASD.

They also worked out on DNMT3B  with reported polymorphism rs1569686 TT, depicting it as risk of CHD in individuals with non-syndromic CHD. These findings may set the future dimensions on new aspects of CHD etiology. 

Authors are asked to respond why they mentioned in this study the data that has been previously reported in their publication " 

Methyltetrahydrofolate-homocysteine methyltransferase reductase gene and congenital heart defects in Down syndrome" 
  • June 2020
  • Genetics & Applications 4(1):12, 

DOI:10.31383/ga.vol4iss1pp12-17

 regarding MTRR gene polymorphism  66A> G (rs1801394) in current manuscript. 

Author Response

Response to Reviewer 3 Comments

We would like to thank the reviewer for valuable comment and observation, which improved our paper. We appreciate the time and effort spent in analyzing our paper. Please find detailed response below.

Point 1: Authors are asked to respond why they mentioned in this study the data that has been previously reported in their publication "Methyltetrahydrofolate-homocysteine methyltransferase reductase gene and congenital heart defects in Down syndrome" June 2020 Genetics & Applications 4(1):12, DOI:10.31383/ga.vol4iss1pp12-17

regarding MTRR gene polymorphism  66A> G (rs1801394) in current manuscript.

Response 1:

the preliminary results of the study, which included 78 individuals with DSCHD+ and 77 with DSCHD-, were presented at the 1st Congress of Geneticists in Bosnia and Herzegovina with international participation in Sarajevo, Bosnia and Herzegovina, October 02-04, 2019. We have cited this publication because the results are similar with the present study, although the number of participants is larger and new groups were included.

Reviewer 4 Report

In this study, the authors investigated the association between MTHFR rs1801133, MTHFR rs1801131, MTRR rs1801394, DNMT1 rs2228611, DNMT3A rs1550117, DNMT3B rs1569686 and DNMT3B rs2424913 gene polymorphisms and congenital heart defects in Down syndrome (DS) individuals. 

The stusy is intresting; however, there is a list of comments that should be enhanced and improved before a positive action is taken:

- The manuscript requires proofreading and revision to improve the quality of English. The authors should enhance the flow of the M, particularly in Introduction (introduction is one paragraph?!?!?).

- Methods of PCR-RFLP (genotyoing) should be elaborated in more details. Likewise, the results.

- Discussion should be strengthened.

- Ethical approval number of this study needs to be added.

Author Response

Response to Reviewer 4 Comments

We would like to thank the reviewer for valuable comments and observations, which improved our paper. We appreciate the time and effort spent in analyzing our paper. Please find detailed response below. All changes in the manuscript are marked up using the “Track Changes” function.

Point 1: The manuscript requires proofreading and revision to improve the quality of English. The authors should enhance the flow of the M, particularly in Introduction (introduction is one paragraph?!?!?).

Response 1: as recommended by the reviewer we will submit the manuscript for revision through the editing services of the MDPI journal and we expanded the introduction with a description of the role of the selected SNPs in folate and homocysteine metabolism, the consequent influence on methylation and the development of CHD.

Point 2: Methods of PCR-RFLP (genotyoing) should be elaborated in more details. Likewise, the results.

Response 2: Methods of PCR-RFLP (genotyping) are elaborated in the supplementary material of the manuscript, we attach Table 1. Supplementary material 1. Primers and PCR-RFLP conditions for the genotyping of DNMT, MTHFR and MTRR genes polymorphisms with their mQTL timepoint.

We apologize and ask the reviewer to give us more precise instructions regarding expanding the results (elaborated in more details).

Table 1. Supplementary material 1. Primers and PCR-RFLP conditions for the genotyping of DNMT, MTHFR and MTRR genes polymorphisms with their mQTL timepoint

Gene Polymorphisms (dbSNP rs)

Position

Forward Primer

Reverse Primer

PCR conditions

PCR

Product

(bp)

Restriction

Endonuclease Enzyme

Reference

mQTL*

timepoint

DNMT1

+32204 A/G

rs2228611

exon

chromosome 19

5’-GTACTGTAAGCACGGTCACCTG-3’

5’-TATGTTGTCCAGGCTCGTCTC-3’

94 °C (5 min)

35x: 94 °C (30s), 55 °C (30s), 72 °C (30s)

72 °C (5 min)

261

BcoDI

[50]

Birth

Childhood

Adolescence

Middle Age

Pregnancy

DNMT3A

-448 A/G

rs1550117

promoter

chromosome 2

5’-ACACACCGCCCTCACCCCTT-3’

5’-TCCAGCAATCCCTGCCCACA-3’

94 °C (5 min)

35x: 94 °C (30s), 55 °C (45s), 72 °C (30s)

72 °C (5 min)

358

HpyCH4III

[51]

-

DNMT3B

-579 G/T

rs1569686

promoter

chromosome 20

5’-GAGGTCTCATTATGCCTAGG-3’

5’-GGGAGCTCACCTTCTAGAAA-3’

94 °C (5 min)

35x: 94 °C (30s), 49 °C (30s), 72 °C (30s)

72 °C (5 min)

225

PvuII-HF

[52]

Birth

Childhood

Adolescence

Middle Age

Pregnancy

DNMT3B

-149 C/T

rs2424913

promoter

chromosome 20

5'-TTGTCCTGAAGCTGGCTACC-3'

5'-ACCAGGAGAGAAGCCAACAG-3'

94 °C (5 min)

35x: 94 °C (30s), 54 °C (30s), 72 °C (30s)

72 °C (5 min)

431

AvrII

[53]

Birth

Childhood

Adolescence

Middle Age

Pregnancy

MTHFR

677C>T

rs1801133

exon

chromosome 1

5’TGAAGGAGAAGGTGTCTGCGGGA3’

5’AGGACGGTGCGGTGAGAGTG3’

94 °C (2 min)

40x: 94 °C (30s), 62 °C (30s), 72 °C (30s)

72 °C (7 min)

198

Hinfl

[54]

Childhood

Adolescence

MTHFR

1298A>C

rs1801131

exon

chromosome 1

5’CTTTGGGGAGCTGAAGGACTACTA3’

5’CACTTTGTGACCATTCCGGTTTG3’

92 °C (2 min)

40x: 92 °C (60s), 56 °C (60s), 72 °C (30s)

72 °C (7 min)

163

Mbo II

[54]

Birth

Childhood

Adolescence

Middle Age

Pregnancy

MTRR

66A>G

rs1801394

exon

chromosome 5

5’CAGGCAAAGGCCATCGCAGAAGACAT3’

5’CACTTCCCAACCAAAATTCTTCAACG3’

92 °C (2 min)

35x: 92 °C (60s), 56 °C (60s), 72 °C (90s)

72 °C (7 min)

150

Afl III

[55]

Birth

Childhood

Adolescence

Middle Age

Pregnancy

DNMT – DNA methyltransferase; PCR – polymerase chain reaction; RFLP – restriction fragment length polymorphism; mQTL – methylation quantitative trait loci

*derived from mQTLdb

Point 3: Discussion should be strengthened.

Response 3:  we apologize and ask the reviewer to give us more precise instructions regarding strengthening the discussion, and whether it is necessary to change the concept of the discussion.

Point 4: Ethical approval number of this study needs to be added.

Response 4: we added the Ethical approval number in the manuscript, previously submitted the ethics commission documentation to the Assistant Editor - revised

Response to Reviewer 4 Comments

We would like to thank the reviewer for valuable comments and observations, which improved our paper. We appreciate the time and effort spent in analyzing our paper. Please find detailed response below. All changes in the manuscript are marked up using the “Track Changes” function.

Point 1: The manuscript requires proofreading and revision to improve the quality of English. The authors should enhance the flow of the M, particularly in Introduction (introduction is one paragraph?!?!?).

Response 1: as recommended by the reviewer we will submit the manuscript for revision through the editing services of the MDPI journal and we expanded the introduction with a description of the role of the selected SNPs in folate and homocysteine metabolism, the consequent influence on methylation and the development of CHD.

Point 2: Methods of PCR-RFLP (genotyoing) should be elaborated in more details. Likewise, the results.

Response 2: Methods of PCR-RFLP (genotyping) are elaborated in the supplementary material of the manuscript, we attach Table 1. Supplementary material 1. Primers and PCR-RFLP conditions for the genotyping of DNMT, MTHFR and MTRR genes polymorphisms with their mQTL timepoint.

We apologize and ask the reviewer to give us more precise instructions regarding expanding the results (elaborated in more details).

Table 1. Supplementary material 1. Primers and PCR-RFLP conditions for the genotyping of DNMT, MTHFR and MTRR genes polymorphisms with their mQTL timepoint

Gene Polymorphisms (dbSNP rs)

Position

Forward Primer

Reverse Primer

PCR conditions

PCR

Product

(bp)

Restriction

Endonuclease Enzyme

Reference

mQTL*

timepoint

DNMT1

+32204 A/G

rs2228611

exon

chromosome 19

5’-GTACTGTAAGCACGGTCACCTG-3’

5’-TATGTTGTCCAGGCTCGTCTC-3’

94 °C (5 min)

35x: 94 °C (30s), 55 °C (30s), 72 °C (30s)

72 °C (5 min)

261

BcoDI

[50]

Birth

Childhood

Adolescence

Middle Age

Pregnancy

DNMT3A

-448 A/G

rs1550117

promoter

chromosome 2

5’-ACACACCGCCCTCACCCCTT-3’

5’-TCCAGCAATCCCTGCCCACA-3’

94 °C (5 min)

35x: 94 °C (30s), 55 °C (45s), 72 °C (30s)

72 °C (5 min)

358

HpyCH4III

[51]

-

DNMT3B

-579 G/T

rs1569686

promoter

chromosome 20

5’-GAGGTCTCATTATGCCTAGG-3’

5’-GGGAGCTCACCTTCTAGAAA-3’

94 °C (5 min)

35x: 94 °C (30s), 49 °C (30s), 72 °C (30s)

72 °C (5 min)

225

PvuII-HF

[52]

Birth

Childhood

Adolescence

Middle Age

Pregnancy

DNMT3B

-149 C/T

rs2424913

promoter

chromosome 20

5'-TTGTCCTGAAGCTGGCTACC-3'

5'-ACCAGGAGAGAAGCCAACAG-3'

94 °C (5 min)

35x: 94 °C (30s), 54 °C (30s), 72 °C (30s)

72 °C (5 min)

431

AvrII

[53]

Birth

Childhood

Adolescence

Middle Age

Pregnancy

MTHFR

677C>T

rs1801133

exon

chromosome 1

5’TGAAGGAGAAGGTGTCTGCGGGA3’

5’AGGACGGTGCGGTGAGAGTG3’

94 °C (2 min)

40x: 94 °C (30s), 62 °C (30s), 72 °C (30s)

72 °C (7 min)

198

Hinfl

[54]

Childhood

Adolescence

MTHFR

1298A>C

rs1801131

exon

chromosome 1

5’CTTTGGGGAGCTGAAGGACTACTA3’

5’CACTTTGTGACCATTCCGGTTTG3’

92 °C (2 min)

40x: 92 °C (60s), 56 °C (60s), 72 °C (30s)

72 °C (7 min)

163

Mbo II

[54]

Birth

Childhood

Adolescence

Middle Age

Pregnancy

MTRR

66A>G

rs1801394

exon

chromosome 5

5’CAGGCAAAGGCCATCGCAGAAGACAT3’

5’CACTTCCCAACCAAAATTCTTCAACG3’

92 °C (2 min)

35x: 92 °C (60s), 56 °C (60s), 72 °C (90s)

72 °C (7 min)

150

Afl III

[55]

Birth

Childhood

Adolescence

Middle Age

Pregnancy

DNMT – DNA methyltransferase; PCR – polymerase chain reaction; RFLP – restriction fragment length polymorphism; mQTL – methylation quantitative trait loci

*derived from mQTLdb

Point 3: Discussion should be strengthened.

Response 3:  we apologize and ask the reviewer to give us more precise instructions regarding strengthening the discussion, and whether it is necessary to change the concept of the discussion.

Point 4: Ethical approval number of this study needs to be added.

Response 4: we added the Ethical approval number in the manuscript, previously submitted the ethics commission documentation to the Assistant Editor - revised

Response to Reviewer 4 Comments

We would like to thank the reviewer for valuable comments and observations, which improved our paper. We appreciate the time and effort spent in analyzing our paper. Please find detailed response below. All changes in the manuscript are marked up using the “Track Changes” function.

Point 1: The manuscript requires proofreading and revision to improve the quality of English. The authors should enhance the flow of the M, particularly in Introduction (introduction is one paragraph?!?!?).

Response 1: as recommended by the reviewer we will submit the manuscript for revision through the editing services of the MDPI journal and we expanded the introduction with a description of the role of the selected SNPs in folate and homocysteine metabolism, the consequent influence on methylation and the development of CHD.

Point 2: Methods of PCR-RFLP (genotyoing) should be elaborated in more details. Likewise, the results.

Response 2: Methods of PCR-RFLP (genotyping) are elaborated in the supplementary material of the manuscript, we attach Table 1. Supplementary material 1. Primers and PCR-RFLP conditions for the genotyping of DNMT, MTHFR and MTRR genes polymorphisms with their mQTL timepoint.

We apologize and ask the reviewer to give us more precise instructions regarding expanding the results (elaborated in more details).

Table 1. Supplementary material 1. Primers and PCR-RFLP conditions for the genotyping of DNMT, MTHFR and MTRR genes polymorphisms with their mQTL timepoint

Gene Polymorphisms (dbSNP rs)

Position

Forward Primer

Reverse Primer

PCR conditions

PCR

Product

(bp)

Restriction

Endonuclease Enzyme

Reference

mQTL*

timepoint

DNMT1

+32204 A/G

rs2228611

exon

chromosome 19

5’-GTACTGTAAGCACGGTCACCTG-3’

5’-TATGTTGTCCAGGCTCGTCTC-3’

94 °C (5 min)

35x: 94 °C (30s), 55 °C (30s), 72 °C (30s)

72 °C (5 min)

261

BcoDI

[50]

Birth

Childhood

Adolescence

Middle Age

Pregnancy

DNMT3A

-448 A/G

rs1550117

promoter

chromosome 2

5’-ACACACCGCCCTCACCCCTT-3’

5’-TCCAGCAATCCCTGCCCACA-3’

94 °C (5 min)

35x: 94 °C (30s), 55 °C (45s), 72 °C (30s)

72 °C (5 min)

358

HpyCH4III

[51]

-

DNMT3B

-579 G/T

rs1569686

promoter

chromosome 20

5’-GAGGTCTCATTATGCCTAGG-3’

5’-GGGAGCTCACCTTCTAGAAA-3’

94 °C (5 min)

35x: 94 °C (30s), 49 °C (30s), 72 °C (30s)

72 °C (5 min)

225

PvuII-HF

[52]

Birth

Childhood

Adolescence

Middle Age

Pregnancy

DNMT3B

-149 C/T

rs2424913

promoter

chromosome 20

5'-TTGTCCTGAAGCTGGCTACC-3'

5'-ACCAGGAGAGAAGCCAACAG-3'

94 °C (5 min)

35x: 94 °C (30s), 54 °C (30s), 72 °C (30s)

72 °C (5 min)

431

AvrII

[53]

Birth

Childhood

Adolescence

Middle Age

Pregnancy

MTHFR

677C>T

rs1801133

exon

chromosome 1

5’TGAAGGAGAAGGTGTCTGCGGGA3’

5’AGGACGGTGCGGTGAGAGTG3’

94 °C (2 min)

40x: 94 °C (30s), 62 °C (30s), 72 °C (30s)

72 °C (7 min)

198

Hinfl

[54]

Childhood

Adolescence

MTHFR

1298A>C

rs1801131

exon

chromosome 1

5’CTTTGGGGAGCTGAAGGACTACTA3’

5’CACTTTGTGACCATTCCGGTTTG3’

92 °C (2 min)

40x: 92 °C (60s), 56 °C (60s), 72 °C (30s)

72 °C (7 min)

163

Mbo II

[54]

Birth

Childhood

Adolescence

Middle Age

Pregnancy

MTRR

66A>G

rs1801394

exon

chromosome 5

5’CAGGCAAAGGCCATCGCAGAAGACAT3’

5’CACTTCCCAACCAAAATTCTTCAACG3’

92 °C (2 min)

35x: 92 °C (60s), 56 °C (60s), 72 °C (90s)

72 °C (7 min)

150

Afl III

[55]

Birth

Childhood

Adolescence

Middle Age

Pregnancy

DNMT – DNA methyltransferase; PCR – polymerase chain reaction; RFLP – restriction fragment length polymorphism; mQTL – methylation quantitative trait loci

*derived from mQTLdb

Point 3: Discussion should be strengthened.

Response 3:  we apologize and ask the reviewer to give us more precise instructions regarding strengthening the discussion, and whether it is necessary to change the concept of the discussion.

Point 4: Ethical approval number of this study needs to be added.

Response 4: we added the Ethical approval number in the manuscript, previously submitted the ethics commission documentation to the Assistant Editor - revised
